

# Insular holobionts: persistence and seasonal plasticity of the Balearic wall lizard (*Podarcis lilfordi)* gut microbiota

Laura Baldo[1,2], Giacomo Tavecchia[3], Andreu Rotger[3], José Manuel Igual[3] and Joan Lluís Riera[1]

[1] Department of Evolutionary Biology, Ecology and Environmental Sciences, University of Barcelona, Barcelona, Spain
[2] Institute for Research on Biodiversity (IRBio), Barcelona, Spain
[3] Animal Demography and Ecology Unit, IMEDEA, Consejo Superior de Investigaciones Científicas, Esporles, Spain

## ABSTRACT

**Background**. Integrative studies of animals and associated microbial assemblages (*i.e.*, the holobiont) are rapidly changing our perspectives on organismal ecology and evolution. Insular vertebrates provide ideal natural systems to understand patterns of host-gut microbiota coevolution, the resilience and plasticity these microbial communities over temporal and spatial scales, and ultimately their role in the host ecological adaptation.

**Methods**. Here we used the endemic Balearic wall lizard *Podarcis lilfordi* to dissect the drivers of the microbial diversity within and across host allopatric populations/islets. By focusing on three extensively studied populations/islets of Mallorca (Spain) and fecal sampling from individually identified lizards along two years (both in spring and autumn), we sorted out the effect of islet, sex, life stage, year and season on the microbiota composition. We further related microbiota diversity to host genetics, trophic ecology and expected annual metabolic changes.

**Results**. All the three populations showed a remarkable conservation of the major microbial taxonomic profile, while carrying their unique microbial signature at finer level of taxonomic resolution (Amplicon Sequence Variants (ASVs)). Microbiota distances across populations were compatible with both host genetics (based on microsatellites) and trophic niche distances (based on stable isotopes and fecal content). Within populations, a large proportion of ASVs (30–50%) were recurrently found along the four sampling dates. The microbial diversity was strongly marked by seasonality, with no sex effect and a marginal life stage and annual effect. The microbiota showed seasonal fluctuations along the two sampled years, primarily due to changes in the relative abundances of fermentative bacteria (mostly families Lachnospiraceae and Ruminococcaceae), without any major compositional turnover.

**Conclusions**. These results support a large resilience of the major compositional aspects of the *P. lilfordi* gut microbiota over the short-term evolutionary divergence of their host allopatric populations (<10,000 years), but also indicate an undergoing process of parallel diversification of the both host and associated gut microbes. Predictable seasonal dynamics in microbiota diversity suggests a role of microbiota plasticity in the lizards' metabolic adaptation to their resource-constrained insular environments.

Corresponding author
Laura Baldo, baldo.laura@ub.edu

Overall, our study supports the need for longitudinal and integrative studies of host and associated microbes in natural systems.

# INTRODUCTION

All organisms live in symbiosis with complex gut microbial communities, which are known to affect a multitude of biological functions (*Levin et al., 2021*), including the host immune response (*Thaiss et al., 2016*), development (*Warne, Kirschman & Zeglin, 2019*) behavior (*Rowe et al., 2020*), thermal regulation (*Moeller et al., 2020*; *Huus & Ley, 2021*), trophic niche preferences and amplitude (*Kohl et al., 2014*; *Kohl et al., 2016*); Leitão-Gonç (*Leitão-Gonçalves et al., 2017*), and the overall efficiency in resource use (*Lindsay, Metcalfe & Llewellyn, 2020*). Collectively this indicates a critical role of the gut microbial communities in forging the host ecology and influencing its evolutionary outcomes (*Shapira, 2016*; *Alberdi et al., 2016*). Extensive work has been done to sort out the relative contribution of the multitude of factors that shape the structure of these communities, revealing a major role of the host genetics, life stage, diet (*Youngblut et al., 2019*; *Rojas et al., 2021*), and geography (*Montoya-Ciriaco et al., 2020*; *Levin et al., 2021*), with important temporal dynamics (*Guo et al., 2021*). These results vary between captive and wild samples (*Youngblut et al., 2019*; *Eliades et al., 2021*) and largely depend on the taxonomic scale of observation, both for microbes (from strain to phylum) and hosts (from individuals up to family level) (*Alberdi et al., 2021*; *Rojas et al., 2021*).

Current theoretical frameworks for microbiome studies indicate the need of natural systems and a population-level approach to address critical eco-evo aspects of gut microbiota-host symbiosis (*Nyholm et al., 2020*; *Alberdi et al., 2021*), including the extent at which these communities are specifics to their hosts (*Mallott & Amato, 2021*), their levels of persistence over time (*Robinson, Bohannan & Britton, 2019*) and their degree of plasticity in response to external factors (*Alberdi et al., 2016*; *Levin et al., 2021*; *Henry et al., 2021*). Studies of temporal variability of the gut microbiota are particularly critical to reach a comprehensive understanding of the diversity of these communities, as well as to shed light into their adaptive potential. Recent long-term studies in vertebrates have shown that the gut microbiota can be highly plastic (*Alberdi et al., 2016*; *Gomez et al., 2019*; *Buglione et al., 2022*), with seasonal fluctuations in response to the host's physiological adjustments and dietary changes over time (*Amato et al., 2015*; *Smits et al., 2017*; *Hicks et al., 2018*; *Baniel et al., 2021*; *Guo et al., 2021*; *Huang et al., 2022*). This suggests a microbial role in buffering the host metabolic needs, which can effectively boost its ecological adaptation by an increase in phenotypic plasticity (*Huang et al., 2022*). Yet, unlike single time point studies, long-term population-level studies of gut microbiota in wild animals remain particularly rare, due to inherent difficulties in individual data collection and demographic monitoring.

Terrestrial vertebrate populations found in small islands provide simplified systems to study local adaptation and phenotypic diversity due to their isolated nature (with dispersal limitations), relatively small sizes, relaxed interspecific interactions (such as predation and competitors), but increased intraspecific competition for the limited local resources (*MacArthur & Wilson, 1967*; *Bittkau & Comes, 2005*; *Velo-Antó, Zamudio & Cordero-Rivera, 2012*). Despite the large interest on insular systems as promoters of evolutionary divergence and ecological speciation, theoretical and empirical studies have rarely targeted the coevolution of the host-associated microbial communities in islands and the potential contribution of this symbiosis to the host insular adaptation (*Lankau, Hong & MacKie, 2012*; *Baldo et al., 2018*; *Davison et al., 2018*; *Michel et al., 2018*; *Buglione et al., 2022*).

The Balearic wall lizard *Podarcis lilfordi,* also known as the Lilford's wall lizard, represents a particularly suitable system for this purpose (*Baldo et al., 2018*; *Alemany et al., 2022a*). The species is endemic to the Balearic Islands and currently comprises several island populations in the archipelagos of Cabrera, Mallorca and Menorca (*Salvador, 2009*). During the last ice age, the ancestral populations present in the mainland of Mallorca and Menorca dispersed to offshore islets and, following a sea level rise, populations remained confined (*Brown et al., 2008*; *Terrasa et al., 2009*), while Mallorca and Menorca ancestral populations were driven to extinction by the introduction of predators (*Alcover, 2000*). All extant populations are bonded by their historical legacy (recent ancestry, ~12,000 years) (*Brown et al., 2008*; *Terrasa et al., 2009*; *Buades et al., 2013*; *Pérez-Cembranos et al., 2020*), while representing independent evolutionary units, with evidence of an ongoing process of diversification (*Pérez-Cembranos et al., 2020*). Their shared climate, reduced area and low biotic diversity make these populations largely controllable and comparable systems, facilitating the study of the major common factors forging and maintaining these lizards' association with gut microbes (*Baldo et al., 2018*; *Alemany et al., 2022a*).

Previous studies have revealed a large conservation of the *P. lilfordi* gut microbiota taxonomic profiles (resembling the typical vertebrate microbiota) (*Baldo et al., 2018*; *Alemany et al., 2022a*), with a potential impact of the phylogeographic history and ecological drift in shaping microbial diversity across different populations (*Baldo et al., 2018*). A recent attempt was made to understand the effect of seasonality and sex in these lizards' microbiota (*Alemany et al., 2022a*), although the lack of individual-level data (specimens from individual populations were pooled) did not allow for a statistically meaningful analysis of population-scale level drivers of microbiota diversity, nor to explore the stability and temporal dynamics of this symbiosis.

In the present study we undertake the first fine population-scale level study of these insular lizards and their gut microbiota by focusing on three well-studied islets/populations from Mallorca (Fig. 1). These sister populations are under a long-term demographic study since 2010: every spring and autumn, individuals are sampled through a capture and recapture method, photo-identified by digitally recording the pattern of ventral scales (*Moya et al., 2015*), sexed and measured (*i.e.,* body length and weight). This has provided unprecedented individual-level and longitudinal data for each population, revealing that, despite their geographic proximity (less than 5 km apart), and similar trophic ecology

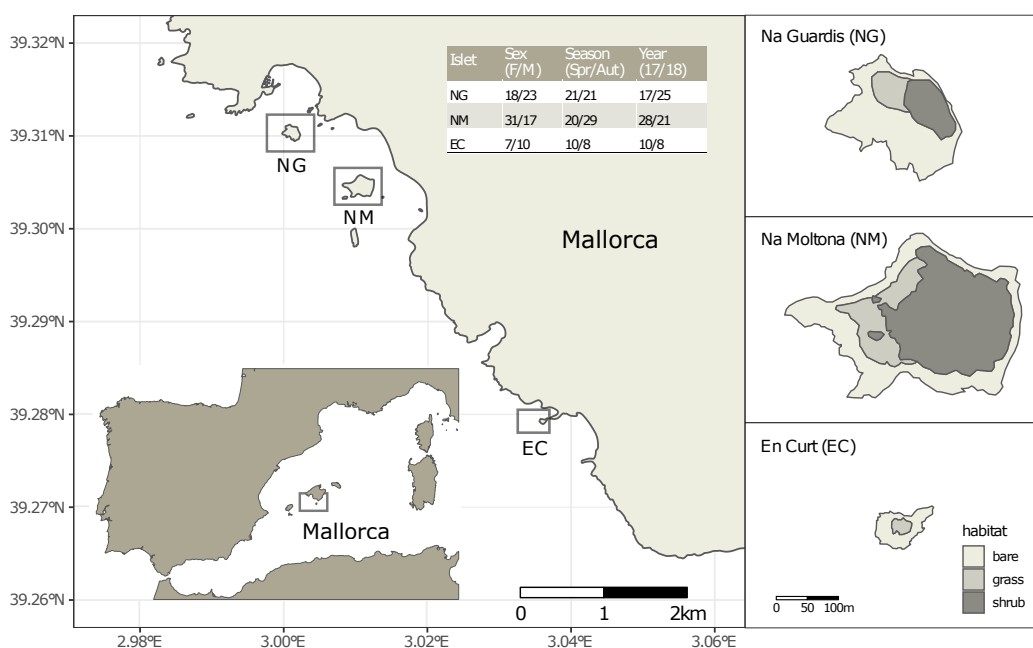

| Islet | Sex (F/M) | Season (Spr/Aut) | Year (17/18) |
|-------|-----------|------------------|--------------|
| NG | 18/23 | 21/21 | 17/25 |
| NM | 31/17 | 20/29 | 28/21 |
| EC | 7/10 | 10/8 | 10/8 |

**Figure 1  Location of the three islets under study and sample statistics.** The inserted table lists the number of fecal samples per sex (females and males, without including one unsexed juvenile per islet), season (spring and autumn) and year (2017 and 2018). See Table S1 for sample metadata.

(*Santamaría et al., 2019*), the three populations differ in demographic parameters and life history traits, such as body growth rate, fecundity, survival, and density (*Rotger et al., 2016*; *Rotger et al., 2021*; *Rotger, Igual & Tavecchia, 2020*). Their diet is primarily based on arthropods, and integrated with mollusks and plant items (nectar, pollen, and seeds) during summer and autumn seasons, when arthropods are scarce (*Santamaría et al., 2019*). The seasonality and extension of the trophic niche (including plant consumption) provide a critical adaptation to the limited and fluctuating island resources (*Pérez-Cembranos, León & Pérez-Mellado, 2016*; *Santamaría et al., 2019*; *Alemany et al., 2022b*).

Taking advantage of this extensive population background here we addressed patterns of the Lilford's wall lizard gut microbiota variation among and within these three allopatric populations over a two-year time. To this purpose we characterized the fecal microbiota from a subset of individually identified lizards for each population, sampled during both spring and autumn seasons. Microbiota data was coupled with extensive metadata to specifically surveyed the impact of the islet phylogeographic distance (as inferred by microsatellites), intrinsic host factors (sex and life stage) and trophic niche (according to stable isotopes data and published fecal content (*Santamaría et al., 2019*) in shaping the gut microbial communties. To shed light on the stability of this association and its potential contribution to lizard insular adaptation, we explored the persistence of microbial taxa and the degree of plasticity of the gut community structure within a population along seasons/years. As these lizard populations experience a strong limitation in resources, particularly critical during the dry autumn season (*Santamaría et al., 2019*), evidence of

microbiota seasonal plasticity might hint to a microbial functional role in buffering the lizard's annual shifts in metabolic needs.

## MATERIALS & METHODS

### Microbiota sampling and metadata collection

Faeces were collected from a total of 109 individually identified specimens of *P. lilfordi* in three islets off the south shore of Mallorca Island: Na Guardis (NG) (42 samples), Na Moltona (NM) (49) and En Curt (EC) (18) (Fig. 1 and Table S1 for sample metadata). The lizard represents the major vertebrate species on each islet, with density ranging from 350 to 2500 ind/ha (*Rotger et al., 2016*).

Sampling was performed during spring (April) and autumn (October) in 2017 and 2018, for a total of four sampling dates ("Spring-17" (37 samples), "Autumn-17" (18), "Spring-18" (14) and "Autumn-18" (40), each referred simply as to "Date"). The reproduction period is extended from Spring to the end of the Summer. Animals begin to lay eggs in May and females can lay two to three clutches, usually of two to four eggs (*Rotger, Igual & Tavecchia, 2020*). All specimens were caught in georeferenced pitfall traps containing sterile fruit juices placed along paths and vegetation edges. Specimens were weighted, and body size measured as snout to vent length (SVL). Life stage (adults, subadults and juveniles) was assigned based on the SVL, according to the mean values for the smallest described subspecies (*Salvador, 2009*). Age of sexual maturity is reached between 1 and 1.5 years old (juveniles are <1 year old) (*Rotger et al., 2016*; *Rotger, Igual & Tavecchia, 2020*). Individuals were sexed by inspection of femoral pores and counting of row ventral scales (males are larger than females and show pores with visible lipophilic compounds (*Salvador, 2009*). Individual-based data of chest images were taken for all individuals and analyzed through the APHIS program for specimen identification and confirmation of sex (*Moya et al., 2015*). After gentle pressing of the specimen abdomen, fecal drops were collected from the cloaca directly into a sterile 2 ml tube filled with 100% ethanol. Specimens were immediately released at their capture point. Samples were placed at −20 °C within the first 24 h from collection and kept refrigerated until processing. Sample preservation in 95–100% ethanol was shown to be effective in maintaining microbial community composition, even for storage up to one week at room temperature (*Song et al., 2016*).

Stable isotopes analysis was conducted at the Laboratorio de Isótopos Estables (LIE-EBD/CSIC, Spain) on a subset of blood samples collected from 71 specimens in spring 2016. Following a 1–2 cm tail cut, drops of blood were immediately collected in capillaries, preserved in ethanol 70% and stored at −20 °C. Samples were dried for 48 h at 60 °C and analyzed before combustion at 1020 °C using a continuous flow isotope-ratio mass spectrometry system. The isotopic composition is reported in the conventional delta ($\delta$) per mil notation (‰), relative to Vienna Pee Dee Belemnite ($\delta^{13}C$) and atmospheric $N_2$ ($\delta^{15}N$).

The species is currently listed as endangered according to the IUCN red list. Specimen sampling and manipulation were carried out in accordance with the ethics guidelines and recommendations of the Species Protection Service (Department of Agriculture,

Environment and Territory, Government of the Balearic Islands), under annual permits given to GT.

## DNA extractions and 16S rRNA Illumina sequencing

Fecal samples were briefly centrifuged, ethanol removed, and the pellet used for DNA extractions with the DNAeasy Powersoil kit (Qiagen), following the manufacturer's protocol. Samples were homogenized with 0.1 mm glass beads at 5,500 rpm, 2 ×45 s using a Precellys Evolution instrument (Bertin Technologies). DNA quality was assessed with Nanodrop and sent to the Centre for Genomic Regulation (CRG) in Barcelona (Spain) for amplicon generation and sequencing. The region V3-V4 of 16S rRNA was amplified using a pool of five forward and reverse primers (including a frameshift to increase diversity) with Nextera overhangs (Table S2). For each sample, amplicons were generated in three-replicates using KAPA Hifi DNA polymerase (Roche), with a first round of PCR (25 cycles); amplicons were then pooled and a 5 µl purified aliquot was used to seed the second PCR (8 cycles) for individual barcoding. Two negative controls (water only) and two mock communities (HM-277D and HM-276D from BEI Resources) were processed along with sample DNA. Barcoded amplicons were pooled at equimolar concentrations and the final library cleaned with the Sequal kit (Invitrogen). The library was sequenced on Illumina MiSeq v3 (600-cycle cartridge, 300 paired-end reads). The final sample dataset did not include any recaptured specimens.

## Amplicon sequence analyses

Demultiplexed sequences were input into Qiime2 (*Caporaso et al., 2010*), primers were removed, and reads were joined with "join-pairs" and filtered with "quality-filter q-score-joined". Sequences were denoised with DEBLUR version 1.1.0 (trim-length = 400, min-reads = 5) (*Amir et al., 2017*) to produce Amplicon Sequence Variants (ASVs). Taxonomic assignment was performed on a trained classifier using the Greengenes database version 13_5 [43]. ASVs classified as mitochondria and chloroplasts or present in the controls (water and mock communities) were discarded (Table S3). Sequences were aligned with Mafft in Qiime2, and hypervariable regions masked. Columns with gaps present in more than 50% of the sequences were removed using trimal (*Capella-Gutiérrez, Silla-Martínez & Gabaldón, 2009*). A rooted phylogenetic tree was built with FastTree (*Price, Dehal & Arkin, 2009*) and used for the unweighted Unifrac analysis. To limit bias in sample sequencing effort, data was rarefied to the minimum sample size (26,003 sequences) and imported into the R environment using the phyloseq package (*McMurdie & Holmes, 2013*).

## Taxonomic composition and diversity

Taxonomic barplots were built with *ggplot* function in the ggplot2 R package.

Beta diversity was visually explored with principal coordinates analysis (PCoA) on Bray-Curtis distances calculated from square root transformed ASV rarefied data using function *cmdscale* in the R stats package. This distance was made euclidean by taking the square root before analysis.

Differences in microbiota composition according to islet, sex, life stage, season, and year were assessed with permutational multivariate analysis of variance (PERMANOVA)

on the same distance matrix after checking for homogeneity in multivariate dispersion. Model selection was performed by first fitting a model with all main terms and all two-way interactions, then refitting the model without the interaction terms with large $p$-values ($p > 0.1$) in the full model based on marginal tests with 10000 permutations. Unlike sequential tests, marginal tests evaluate each term against a model containing all other terms. Therefore, the refitted model contains tests for the chosen interactions and for the main terms that do not form part of an interaction term. PERMANOVA was done with function *anova2*, and multivariate homogeneity in dispersions with function *betadisper*, both in the R package 'vegan' (*Oksanen et al., 2020*).

Alpha diversity was evaluated using the Chao1 richness estimator and the Shannon diversity index on seasonal datasets using the function *plot_richness* in phyloseq. Differences by islet and season were tested with two-way analysis of variance (ANOVA) models with marginal tests (Type II sums of squares) due to unbalanced design. *Post-hoc* pairwise comparisons were done with the 'emmeans' R package (*Lenth, 2022*).

### Islets microbial markers

Bacteria taxa driving differences in microbiota composition across populations (*i.e.,* islet biomarkers) were searched through a double approach: the Dufrene-Legendre Indicator Species analysis using the *indval* function in the labdvs R package (*Roberts, 2019*) and the Linear discriminant analysis Effect Size (LEfSe) for biomarker discovery (*Segata et al., 2011*). Both approaches retrieve differences across pairs of groups considering both presence-absence and differential abundances. Results obtained from the two methods were intersected to account for potential methodological biases (*Nearing et al., 2022*).

For both approaches, the input dataset was rarefied, retaining only ASVs with more than 100 total sum counts to reduce sparsity issues (*Nearing et al., 2022*), and the dataset split into spring and autumn samples, performing the analysis on seasonal datasets. This allows to retrieve only markers that are islet-specific but season-independent. Juveniles were excluded from this analysis due to insufficient representation in the sample. *Indval* analyses were run on all taxonomic levels (from ASV to phylum), binning counts with the function *aggregate* in R stats package. Significant ASVs/taxa were retained when relfreq $\geq$ 0.6 (minimum relative frequency of occurrence within a population for ASV/taxa to be retained) and $p < 0.01$. Results from spring and autumn datasets were crossed to obtain season-independent discriminatory features. LEfSe analyses were run in the Galaxy web application setting the class to "Islet" (Kruskal-Wallis among classes $p = 0.01$, and pairwise Wilcoxon test between subclasses, $p = 0.01$), and a threshold on the logarithmic LDA set to three, with one-against-all strategy. The analyses were run on ASV and higher taxa levels separately. Only ASV/taxa retrieved in both seasonal datasets were retained as islet biomarkers.

### Microbiota and host genetic and trophic distances

Microbiota distances among islets were calculated as the islet centroids computed from the Bray-Curtis and unweighted Unifrac distance matrices using the function *distance* in the phyloseq package and function *dist_between_centroids* in the usedist package (*Bittinger,*

*2020*). To take into account intrapopulation microbiota variance in centroid estimates, multiple distance matrices were built on distinct core datasets (50 to 90%), where the core is a subset of ASVs shared among a cutoff percentage of individuals within a population. Host genetic distances among the three islets/populations were inferred using average *Fst* distances according to published microsatellites data (*Rotger et al., 2021*).

Differences in mean values of stable isotopes among islets were tested with generalised least squares (GLS) to account for strong heteroskedasticity. *Post-hoc* pairwise comparisons were performed using the Satterthwaite approximation for degrees of freedom and the Tukey method for *p*-value adjustment. Models were fitted with function *gls* in the 'nlme' R package (*Pinheiro et al., 2022*), and pairwise comparisons with the 'emmeans' package (*Lenth, 2022*).

### Persistence of ASVs over sampling dates and seasonal microbial markers

Persistence was assessed as the portion of the microbial ASVs that was consistently retrieved in all four sampling dates, considering only those ASVs that occurred in at least 50% of the specimens within a single date. The four 50% core datasets were then compared to retrieve common ASVs, *i.e.,* the microbial component present along all sampling dates. Comparing 50% cores by date, instead of using the full microbial diversity per date, reduces the probability that only a few specimens per population contributed to the observed pattern. Venn diagrams were produced using the online tool at http://bioinformatics.psb.ugent.be/cgi-bin/liste/Venn/calculate_venn.htpl.

To estimate bacterial features responsible for seasonal differences within a population (*i.e.,* seasonal markers), we undertook a similar double approach, performing both LEfSe and *indval* analyses. The analyses were run on NM and NG, which had large sample representation for both autumn and spring 2017 and 2018 (above 15 samples each season/year). Same season samples within each population were treated as a single group and discriminatory features were estimated as above (for LEfSe analysis, class was set to "Season").

## RESULTS

We sequenced the fecal microbiota of 109 specimens from the three closed populations/islets south of Mallorca (Fig. 1). Fecal samples were associated to four major categorical variables: islet, sex, life stage, season (spring and autumn) and year (2017 and 2018) (see Table S1 for sample metadata). The final microbiota dataset encompassed an even representation of each variable, except for life stage (nearly 80% of the specimens were adults).

After extensive quality filtering and removal of taxa found in the controls (Table S3), we obtained a total of 6195163 sequences and 2313 ASVs (10 minimum reads) (abundance matrix with taxonomy is available at Mendeley Data, DOI: 10.17632/bc5nxsxgxd.1): 1360 ASVs in EC, 1647 in NG and 1677 in NM. Of these ASVs, 91 (EC), 74 (NG) and 70 (NM) were present in at least 80% of the specimens within each islet/population (*i.e.,* they form the islet core microbiota); 24 of these ASVs were common to all islets (*i.e.,* common core

microbiota). According to rarefaction curves, sequencing effort was sufficient to approach the maximum diversity for most samples (Fig. S1). Data was nonetheless rarefied to the minimum sample depth of 26,003 reads (corresponding to 1,933 ASVs) to account for potential bias in sequencing effort and sparsity, and used for all subsequent analyses.

## Highly conserved microbial taxonomic profile among wall Lilford's wall lizard allopatric populations

The overall fecal microbial dataset comprises a total of 18 unique phyla, 36 classes, 64 orders, 94 families, 134 genera, and 66 species. The taxonomic profile of the most abundant taxa was remarkably conserved at phylum, family, and genus level across all individuals (Fig. S2), with no major compositional differences across islets, between males and females and along the four sampling dates (Fig. 2). In all cases, the two most abundant phyla were Firmicutes (43%) and Bacteroidetes (38%), with similar relative abundances, followed by Proteobacteria (8%) and Tenericutes (6%) (Fig. 2). Dominant families were Bacteroidaceae (21%), Lachnospiraceae (15%), Ruminococcaceae (8%) and Porphyromonadaceae (8%). The most abundant genera were *Bacteroides* (21%), *Parabacteroides* (7%), *Anaeroplasma* (5%), *Oscillospira* (4%), *Odoribacter* (3.5%) and *Roseburia* (2.6%) (Fig. 2). Only 8% of the ASVs (154 out of 1,933) reached species classification (confidence threshold 80%); the most abundant species were *Parabacteroides gordonii* (4.4%), *Clostridium ramosum*, *Parabacteroides distasonis* and *Akkermansia muciniphila* (all <1%).

## Islet and season as major variables shaping the microbiota structure

PERMANOVA analysis on the entire dataset (*i.e.,* including all life stages, *i.e.,* juveniles, subadults and adults) indicated statistically significant clustering by the interaction between islet and season ($P \leq 0.0001$), and islet and life stage ($P = 0.005$), marginal differences by year ($P = 0.0449$), and no sex effects ($P = 0.1197$) (Table 1A). Given that juveniles were underrepresented in our dataset ($n = 9$ out of 109 individuals), life stage effect should be taken with caution. When excluding juveniles (leaving subadults and adults), PERMANOVA analysis still supported a strong islet by season interaction ($P \leq 0.0001$) and a marginal year effect ($P = 0.0396$), yet no differences by either sex or life stage (Table 1B). This suggests that differences in life stage were mostly due to the juvenile stage, with no differences between adults and subadults. Therefore, juveniles were excluded from all subsequent analyses.

Principal coordinates analysis showed that EC hosted a clearly distinct microbial community, while NG and NM substantially overlapped on the subspace defined by PCo1 and PCo2 (Fig. 3). In addition, *post-hoc* tests by islet showed that season was a statistically significant factor in every case, but most clearly in NM and NG (NM: $P = 0.0002$, NG: $P \leq 0.0001$, EC: $P = 0.008$) (Fig. 3).

According to alpha diversity analyses on seasonal datasets (Fig. 4), spring showed a highly homogenous pattern of diversity, with no major differences across any islet pairwise (both Chao1 and Shannon, $p > 0.05$), whereas autumn marked a large difference among all three populations, with EC being the most diverse (Chao1 and Shannon, $p < 0.05$ for all islet pairwises, except for NM-EC, $p > 0.05$ Shannon). Within individual populations,

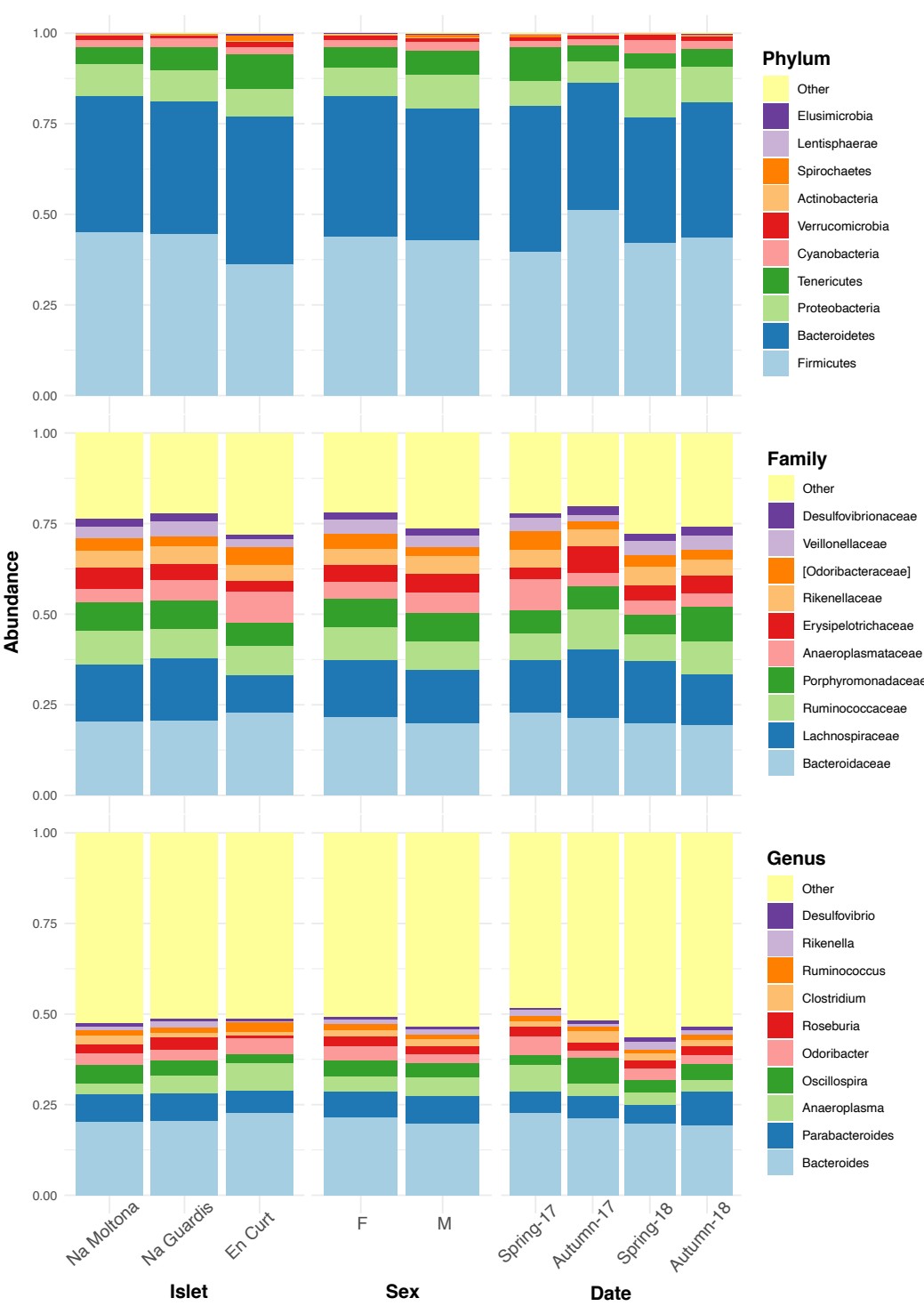

**Figure 2 Taxonomic composition of the *P. lilfordi* gut microbiota at phylum, family, and genus levels according to "islet", "sex" and sampling "date" (season-year).** Juveniles ($n = 9$) were excluded. The legends list only the top 10 taxa (all remaining taxa were included in "Others"). No major taxonomic differences were observed as a function of any of the variables. For individual specimen taxonomic profile, see Fig. S2.

**Table 1** Results of PERMANOVA (9999 permutations).

| | Df | Sum of squares | R² | pseudo F | Pr(>F) |
|---|---|---|---|---|---|
| *a) Entire dataset* | | | | | |
| Sex | 2 | 0.658 | 0.0183 | 1.1061 | 0.1197 |
| Year | 1 | 0.376 | 0.0104 | 1.2641 | 0.0449 |
| Islet:Season | 2 | 0.998 | 0.0277 | 1.6778 | **0.0001** |
| Islet:Life_Stage | 6 | 2.099 | 0.0583 | 1.1759 | **0.0050** |
| Residual | 94 | 27.970 | 0.7760 | | |
| Total | 108 | 36.043 | 1 | | |
| *b) Without juveniles* | | | | | |
| Sex | 1 | 0.309 | 0.0095 | 1.0432 | 0.2876 |
| Life_Stage | 1 | 0.309 | 0.0095 | 1.0433 | 0.2957 |
| Year | 1 | 0.379 | 0.0117 | 1.2818 | 0.0396 |
| Islet:Season | 2 | 1.050 | 0.0323 | 1.7744 | **0.0002** |
| Residual | 91 | 26.921 | 0.8270 | | |
| Total | 99 | 32.552 | 1 | | |

**Notes.**
All tests are marginal. Bold values for $P < 0.01$.

spring showed a richer community in NG ($p < 0.001$ for Chao1, but not significant for Shannon) but not in NM ($p > 0.05$ both indexes), while the opposite pattern was observed in EC, with autumn being most diverse ($p < 0.05$ both indexes). No statistically significant differences in alpha diversity were found between sexes within individual islets ($p > 0.1$ for both indexes).

## Among population microbiota diversity is explained by both lizard phylogeography and trophic niche

To explore the "islet/population" effect on the microbiota diversity (Figs. 3 and 4), we searched for biomarkers of each islet using a double approach (*indval* and LEfSe analyses, see Methods). A total of 14 ASVs and two taxa were retrieved by the two methods, which discriminated across islets according to both autumn and spring datasets (Fig. 5 and Table S4 for results and taxonomic classification). In accordance with the PCoA clustering (Fig. 3), most discriminatory features were enriched in the EC islet and virtually found only on this islet, with no occurrence in either NM or NG (relabund values close to 0 for both NM and NG, Table S4). Only three ASVs were found to be specifically enriched in NM, although not unique to this islet, while NG showed no islet-specific microbial markers. Most discriminatory ASVs belonged to the order Bacteroidales, and fermentative families Bacteroidaceae and Porphyromonadaceae. Their abundance across individuals was highly comparable between spring and autumn, suggesting stability in biomarkers relative abundance over time (Fig. 5). At taxa level, the islet EC showed a unique enrichment in the phylum Elusimicrobia (virtually absent in the other islets) and in the genus *Vibrio*, specifically in the species *Vibrio rumoiensis* (phylum Proteobacteria) (Fig. 5). No specific taxa markers were detected for either NG or NM.

Bray-Curtis microbiota centroid distances among islets, calculated using distinct core subsets per islet (50, 60, 70, 80%), indicated high microbial community relatedness

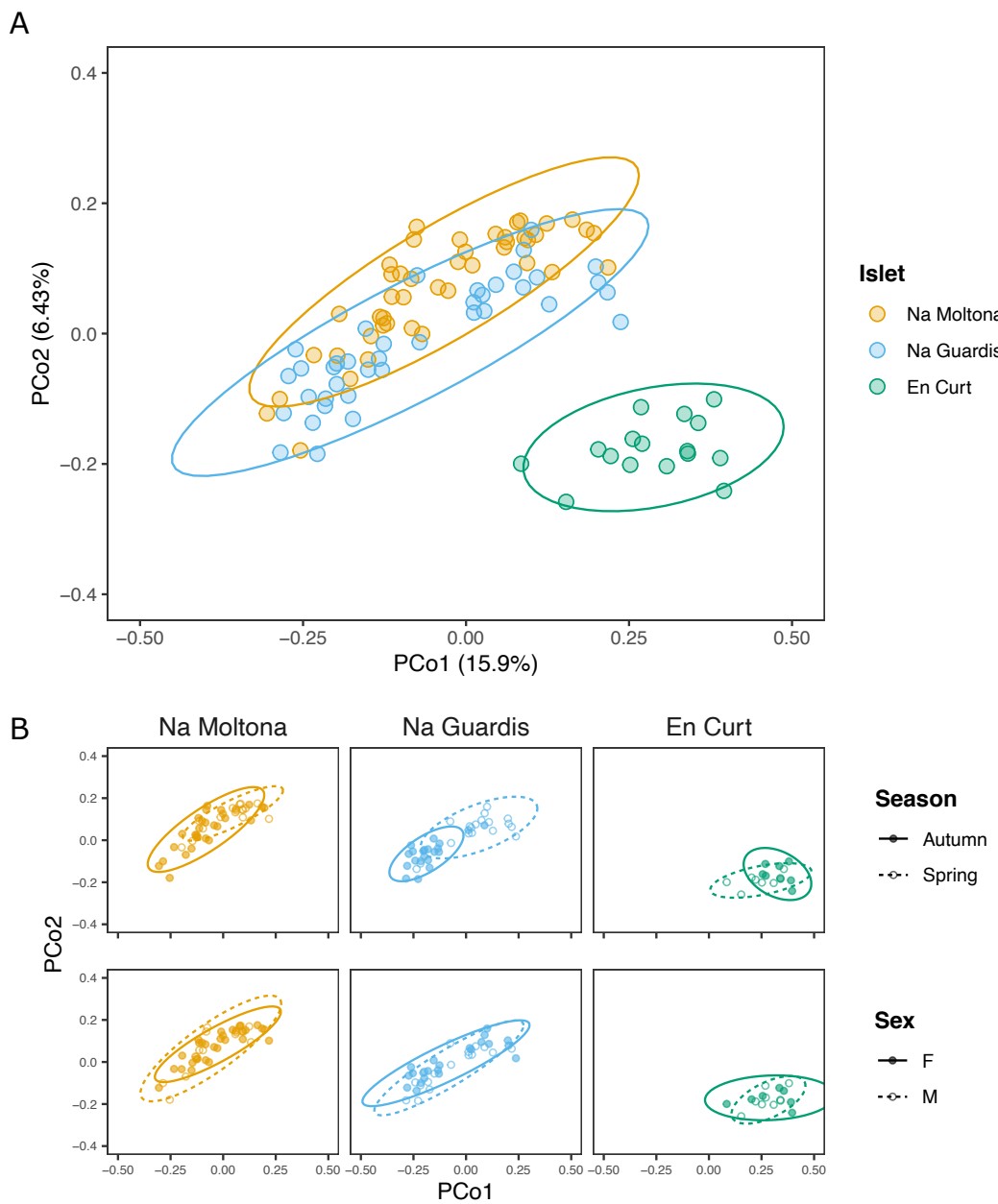

**Figure 3** **PCoA based on Bray-Curtis distances of the *P. lilfordi* gut microbiota depicting diversity (A) among populations ("islets") and (B) within populations, according to "season" and "sex".** Dots represent specimens. Juveniles ($n = 9$) were excluded. Ellipses (calculated with stat ellipse in ggplot2) enclose 95% of the expected values around centroids assuming a $t$ distribution. Data were square root transformed. Microbiota differences were driven by "islet", "season" (within each islet), but not "sex".

between the two largest islets, NG and NM, with EC being the most differentiated (Fig. 6A, see Fig. S3 for Unifrac distances). These microbiota distances were concordant with the host population genetic distances based on previously estimated *Fst* values of microsatellite diversity (*Rotger et al., 2021*).

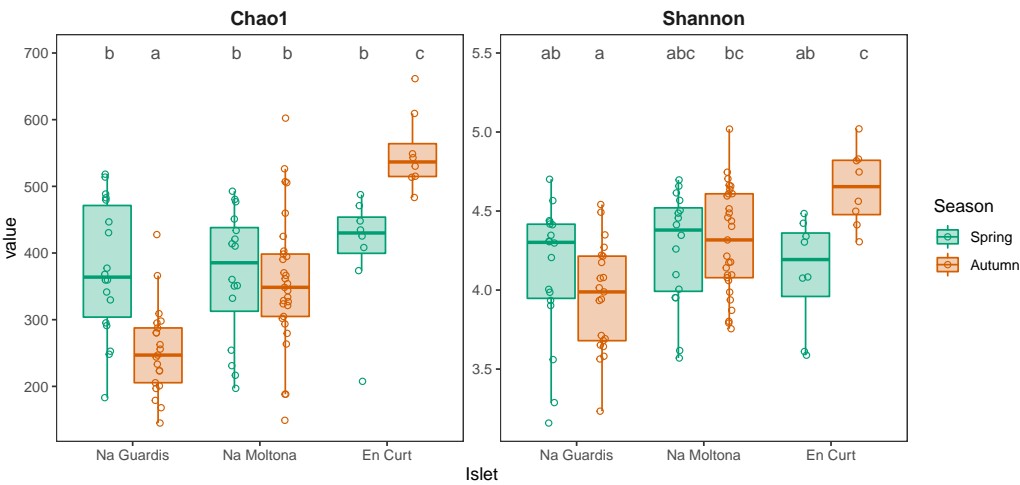

**Figure 4 Alpha diversity of gut microbiota by islet according to Chao1 and Shannon, estimated on seasonal datasets.** Top letters show pairwise statistical differences ($p < 0.05$). Significant differences among islets are observed only for the autumn dataset.

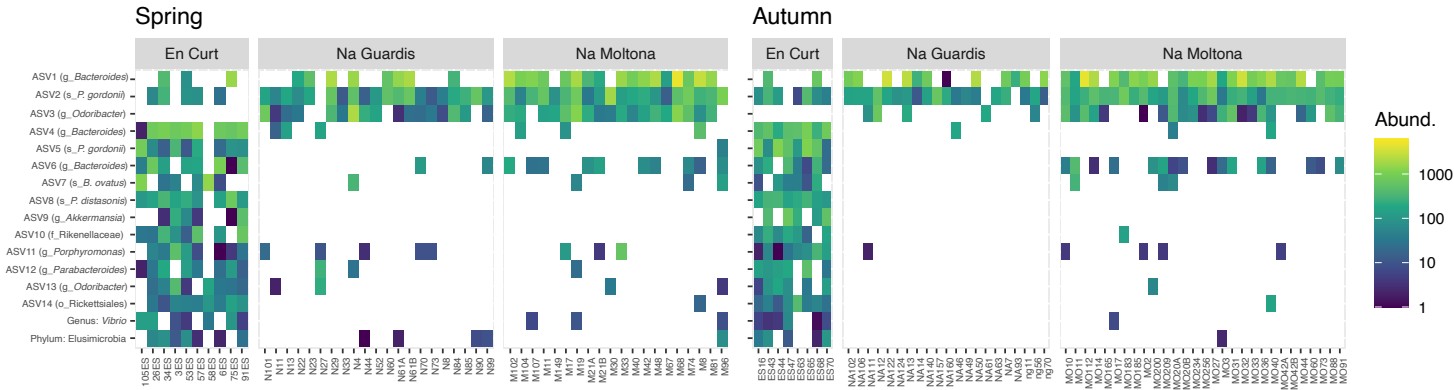

**Figure 5 Heatmap of the microbial markers driving significant differences across islets, consistently in spring and autumn (14 ASVs and two taxa).** Pattern of ASV relative abundance per specimen (x axis, ordered by their mean abundances in spring) is highly concordant between seasons, despite the datasets include different sets of individuals. Heatmaps were built on log-transformed data grouped by islet and season. Data were restricted to ASVs and taxa retrieved by both *indval* and LEfSe approaches (for indavl, relfreq $\geq 0.6$ and $p < 0.01$; for LEfSe LDA $> 3$ and $p < 0.01$). See Table S4 for full taxonomic classification and statistics.

Trophic niche distances among the three populations were investigated through stable isotopes on sample sets from 2016 (data available at Table S5). Findings indicated that both carbon-13 and nitrogen-15 differed among islets (Fig. 6B), while *post-hoc* pairwise analyses showed that in both cases EC displayed higher values of both carbon-13 and particularly of nitrogen-15 with respect to both NG (N-15: t[46.8] = 19.7, $P < 0.001$; C-13: t[37.1] = 7.57, $P < 0.001$), and NM (N-15: t[35.9] = 14.38, $P < 0.001$; C-13: t[41.8] = 3.44, $p = 0.038$), while the latter islets differed for carbon-13 (t[20.1] = 4.01, $P = 0.0019$) but only marginally for nitrogen-15 (t[25.8] = 2.58, $P = 0.0407$).

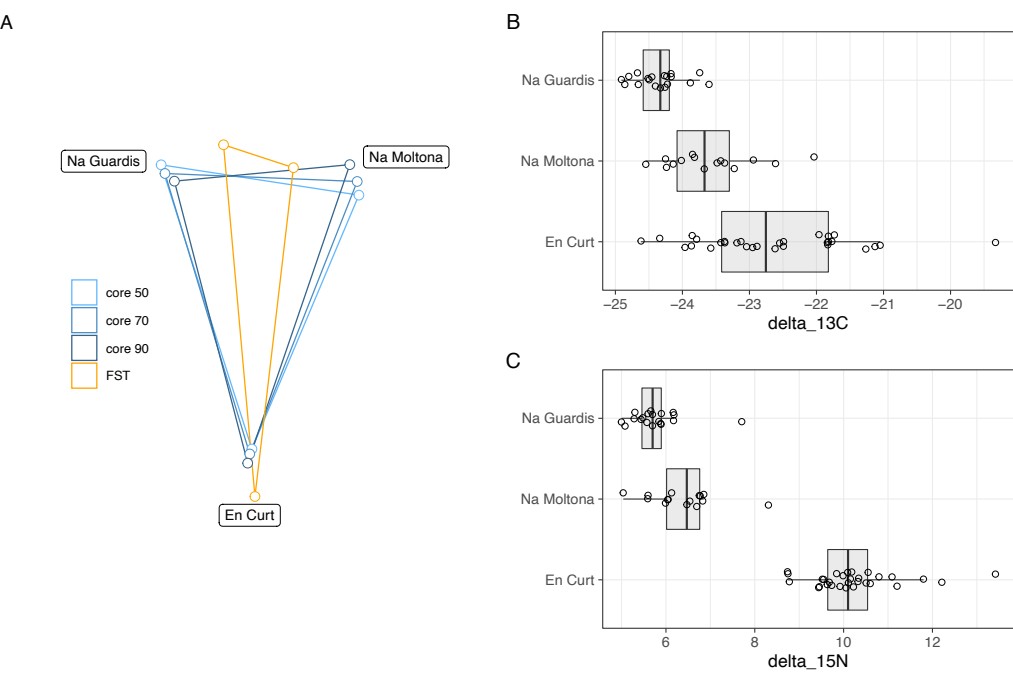

**Figure 6** (A) Host genetic and gut microbiota distances among the three islets/populations and (B–C) host trophic niches based on stable isotopes, carbon-13 (B) and nitrogen-15 (C). (A) Superimposed map of microbiota centroids distances (Bray-Curtis) per islet calculated on different core subsets, and host population genetics distance based on *Fst* values estimated on available microsatellites from a previous study (pairwise *Fst*, EC-NM: 0. 135, EC-NG: 0.144, NM-NG: 0.03, $p = 0.001$ for all pairwises) (*Rotger et al., 2021*). See Fig. S3 for results based on Unifrac distances). (B–C) Stable isotopes were etimated for each population based on a dataset from spring 2016 (Table S5). EC displayed higher values of both carbon-13 and particularly of nitrogen-15, with minor differences among NM and NG. The relative distances among islets according to host genetics, trophic ecology and gut microbiota are highly congruent.

Overall, the microbiota distances were consistent with both the host population genetic and ecological distances.

## Within island/population microbiota diversity: microbiota persistence and seasonal effect

Microbial diversity within islets was primarily driven by "season" ($p = 0.001$, no juveniles) (Table 1 and Fig. 3). To evaluate the level of microbiota plasticity within a population over time (*i.e.,* degree of changes in relative abundance and/or turnover of microbial taxa), we investigated both persistence of microbial ASVs along the four sampling "dates" (spring-17, autumn-17, spring-18 and autumn-18) and enrichment patterns as a function of season (spring *vs* autumn). The analyses were restricted to the two islets with the largest sample representation per date, NM and NG (Fig. 1).

Persistent ASVs across all four sampling dates represented 30.5% (102 ASVs) and 49% (151 ASVs) of the NG and NM core diversity by date, respectively (Fig. S4), with 82 ASVs being common to both islets (Table S6). Taxonomic profiles of these persistent ASVs were largely congruent between islets, with the majority belonging to the family

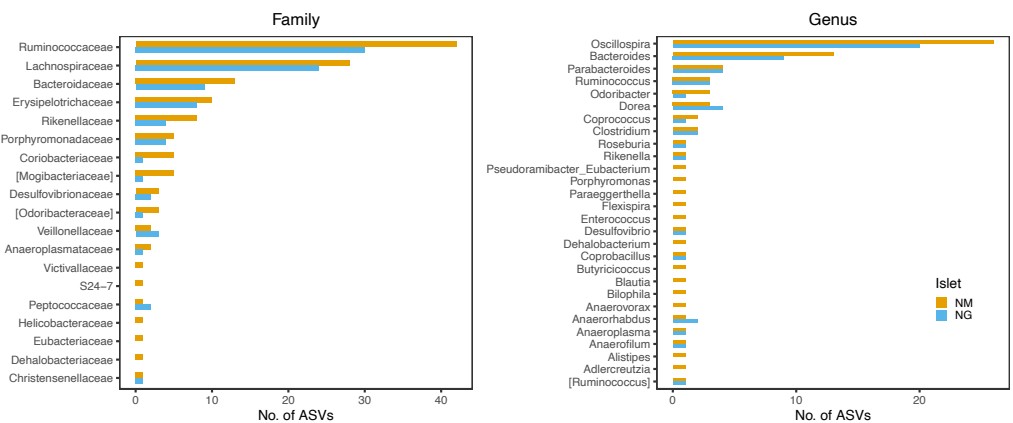

**Figure 7 Family and genus-level taxonomic profile of persistent ASVs retrieved along the four sampling dates.** The bars indicate the frequency of ASV per each taxon. The two islets shared a highly similar taxonomic profile.

Ruminococcaceae and genus *Oscillospira*, followed by members of the Lachnospiraceae and Bacteroidaceae (particularly of the genus *Bacteroides*) (Fig. 7).

According to microbiota centroids by "Date" on Bray-Curtis distances, same season microbiotas sampled in 2017 and 2018 were highly similar and diverged from microbiotas collected in distinct seasons (Fig. 8A), indicating that the microbiota structure alternates across seasons in a quite conserved manner: the microbiota configuration state shifts from spring 2017 to autumn 2017, then goes back to a similar state in spring 2018 and finally shifts again to the autumn configuration in 2018. This pattern was observed in both islets and was robust to the use of different core subsets, up to 90% (Fig. S5), suggesting it is largely driven by quantitative changes in relative abundances of core ASVs.

Enrichment analyses with *indval* and LEfSe analyses identified 63 and 105 ASVs that were differentially abundant between spring and autumn in NM and NG, respectively (of these, 20 ASVs (NG) and 11 ASVs (NM) were consistently retrieved by both methods, Table S7). Most of these ASV seasonal markers (spring and autumn enriched) showed a clear fluctuation in relative abundance along the four sampling dates, for both islets (Fig. 8B). A proportion of them corresponded to persistent ASVs (seven of the 20 ASVs in NG, and five of the 11 in NM), with the majority showing an average abundance different from zero for all dates (see in particular NM autumn-enriched ASVs) (Fig. 8B). Only few ASVs dropped below the threshold of detection during one/two dates (particularly in NG, Autumn-17), while being present on all other dates. The taxonomic profile of enriched ASVs was largely similar between seasons and islets: most ASVs (~70%) belonged to the order Clostridiales and family Ruminococcaceae and Lachnospiraceae (Fig. 8B).

A similar pattern was retrieved for higher taxonomic levels, with most taxa being consistently retrieved across all dates, while fluctuating in relative abundances in both islets (Fig. S6). Notably, few species showed a clear seasonal-associated presence/absence pattern, including the disease-associated species *Enterobacter hormaechei* and *Lawsonia intracellularis* (autumn-specific for NG). While the taxonomic composition of enriched

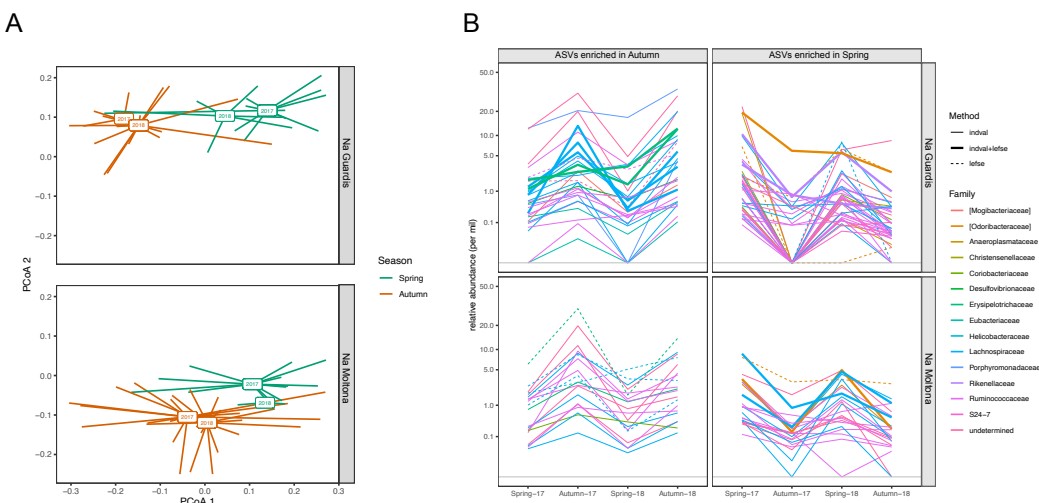

**Figure 8  Compositional dynamics of the gut microbiota in NG and NM islets along the four sampling dates (spring and autumn 2017 and 2018).** (A) PCoA based on Bray-Curtis distances on square root transformed values. Square boxes depict centroids for each year and season, with lines connecting centroids with individual observations. Microbiota configuration changes across seasons in a repetitive manner and consistently in the two populations. Results were robust to the use of different core subsets (see Fig. S5). (B) Variation in mean relative abundance along "dates" of ASVs that were significantly enriched in either spring or autumn according to LEfSe and/or *indval* analyses. A clear pattern of seasonal fluctuations can be observed for most ASVs, with the majority belonging to the families Ruminococcaceae and Lachnospiraceae. A total of 20 ASVs (NG) and 11 ASVs (NM) were consistently retrieved by both methods (see also Table S7).

taxa was overall largely specific to each islet, few taxa displayed a highly congruent pattern between islets, supported by both methods (LEfse and *indval*): the genus *Odoribacter* and family *Odoribacteraceae* (enriched in spring), and the genus *Anaerofilum* (enriched in autumn). Both taxa form part of the core microbiota (present in at least 80% of all specimens).

# DISCUSSION

A fundamental question in the study of the host-microbes symbiosis is to which extent this association is resilient to spatio-temporal changes and what are the processes influencing such (lack of) divergence, which ultimately affects patterns of coevolution in animals (*Mallott & Amato, 2021*). The use of relatively simple natural systems (small insular vertebrates) and a population-level approach with individual-level data are critical to address this question.

Here we provided a first in-depth exploration of population-level drivers of the gut microbiota structure in the Balearic wall lilford's lizard, focusing on the impact of host phylogeographic history, dietary niches, host intrinsic traits (sex and lifestage) and temporal variables (year and season), with the major aim of sheding light on the strength of this symbiotic association, its level of plasticity and putative role in the host insular adaptation.

## Microbiota diversity across islet populations

What drives the early steps in microbiota diversification among populations once the reproductive boundaries are set? And in which aspects do these associated communities start diverging following the host genetics divergence and adaptation to the new environment?

Gut microbiota divergence across host allopatric populations is largely a function of timing since population divergence (*i.e.,* the phylogeographic history), putative adjustments/transitions in the host ecological niche following separation, and exposure to distinct pools of environmental bacteria (*Lankau, Hong & MacKie, 2012*; *Michel et al., 2018*). The relative impact of these processes on microbial community changes will depend on the plasticity of the gut microbiota in response to both host selectivity/filtering (underpinned by the host genetics) (*Alberdi et al., 2016*; *Gomez et al., 2019*), the level of microbial transmission across generations (both vertical and horizontal) and the impact of stochastic events (*i.e.,* ecological drift) (*Lankau, Hong & MacKie, 2012*; *Baldo et al., 2018*; *Michel et al., 2018*).

Here we observed a largely homogeneous taxonomic profile of the *P. lilfordi* gut microbiota among the different allopatric populations, without any major difference according to the studied variables (islet, sex, and sampling date) (Fig. 2 and Fig. S2). All populations present a similar dominance of the phyla Firmicutes and Bacteroidetes, families Lachnospiraceae, Ruminococcaceae and Porphyromonadaceae and genera *Bacteroides* and *Parabacteroides* (Fig. 2). This is largely consistent with our previous study based on microbial content of seven populations of the same species from Menorca, using full intestine tissues (*Baldo et al., 2015*), and a recent study on fecal microbiota including additional five islets from Cabrera and Mallorca (*Alemany et al., 2022a*). This conservatism of the taxonomic profile likely results from common host genetic constraints (*Mallott & Amato, 2021*), exerting a strong imprinting and stabilising effect on the major microbial membership composition, which overcomes the exposure to distinct local environments (*Baldo et al., 2018*; *Rotger et al., 2021*), a pattern that has been previously observed in recently diverged species (*e.g.,* in the Galapagos finch (*Michel et al., 2018*) and cichlid fishes (*Baldo et al., 2017*).

Nonetheless, at finer taxonomic level, *i.e.,* in terms of ASVs, the three islet populations carried their unique microbial signature (Fig. 3). Once excluding the life stage and season effect (therefore working only on adults/subadults and individual seasons), islet was indeed a statistically significant clustering factor, although driven by a limited number of ASVs and taxa, mostly specific to the smallest islets EC (Fig. 4). Interestingly, islet biomarkers had comparable enrichment patterns across the spring and autumn datasets, thus showing independence from a seasonal effect and suggesting they might represent new stable uptakes from the local microbial pool. Whereas the functional role of these islet-specific markers is unclear (such as the unique enrichment in Elusimicrobia found in the smallest islet of EC, a recently identified animal-associated phylum which rely on fermentation (*Méheust et al., 2020*), these markers are important targets of future studies on the role of gut microbiota in host local adaptation.

Microbiota distances among the three populations/islets were consistent with their host population genetic distances, with NG and NM being the genetically and geographically
closest populations, as well as hosting the most similar microbial communities (Figs. 1 and 6A). This pattern is consistent with a phylogeographic scenario of microbiota divergence following their host diversification (Fig. 6A). According to published microsatellites data, NM and NG diverged about 2,000 –4,000 years ago, with EC representing the most distant population (*Rotger et al., 2021*). A putative event of gene flow might have occurred between NM and NG in recent times (<200 years, associated to a single specimen translocation) (*Rotger et al., 2021*), which could have resulted into dispersion of microbial taxa and homogenization of the overall microbial diversity between these two populations. However, these two populations show clear differences in their morphological and life history traits (smaller body size and marked senescence in NG), indicating an ongoing process of divergence (*Rotger, Igual & Tavecchia, 2020*; *Rotger et al., 2021*). The microbiota clustering observed might still be compatible with a process of codivergence of microbes with their hosts, as repeatedly reported in other animal systems (*Lim & Bordenstein, 2020*; *Mallott & Amato, 2021*). In lizards, comparisons of captive *versus* natural populations have indeed proved their ability to transmit microbes across generations, suggesting a possible scenario of retention of ancestral bacteria following the *P. lilfordi* vicariance process (*Baldo et al., 2018*).

The analysis of the trophic niche by stable isotopes provided a seemingly compatible clustering: EC deviates in its trophic niche from both NG and NM, probably due to the rocky nature of this small islet (0.30 ha) (Fig. 1) with a very low biotic index and limited resources, potentially causing metabolic stress and a higher incidence of cannibalism. A wider range of both carbon-13 and nitrogen-15 values for this population (Fig. 5B) also indicate a larger trophic niche breadth, including marine food items obtained along the shore (molluscs and crustaceans), not predominant in either NM or NG (personal observation of behavior). Dietary adjustments can greatly impact the microbiota composition in lizards (*Kohl et al., 2016*; *Jiang et al., 2017*; *Montoya-Ciriaco et al., 2020*; *Buglione et al., 2022*), suggesting that the observed differences in trophic niche can partly drive the observed microbiota clustering among populations.

At present, both the phylogeographic and ecological scenarios are compatible with the observed pattern of islet microbiota divergence, inviting caution in data interpretation and especially in phylosymbiosis claims when ecological aspects are not fully understood. A selection of putatively heritable gut bacteria is currently being analysed at strain level to resolve the evolutionary trajectories of gut microbes in the three populations. Additionally, a more in-depth study of the trophic niche should be undertaken to clarify major ecological/dietary differences across populations.

## Microbiota diversity within populations

Dissecting the drivers of microbial diversity within natural populations presents several challenges due to the multiple concurrent host variables to consider (*e.g.*, sex and life stage), and the spatial–temporal effect (*e.g.*, year and season). To date, most studies on wild animals and lizards in particular have indeed targeted variation between natural populations (*Ren et al., 2016*; *Jiang et al., 2017*; *Zhang et al., 2018*; *Alemany et al., 2022a*),
with only few addressing intrapopulation diversity aspects in natural systems (*Ren et al., 2016*; *Kohl et al., 2017*).

Despite the inherent limitations of working with natural systems, our sampling design was optimized to provide a statistically meaningful dataset for exploring the relative contribution of some of major players in microbiota diversity within populations, *i.e.*, sex, and temporal variables (season and year), while controlling for life stage. Our results showed that the observed diversity within each islet/population was not significantly driven by sex, partly by life stage and year, while being strongly affected by seasonal dynamics (Table 1 and Fig. 3).

In general, lizards in Mediterranean islands are largely omnivorous and opportunistic in trophic behavior, modulating niche width and food preferences along the year in response to both resource availability and energy requirements (affected by reproductive behavior and external temperature) (*Pérez-Cembranos, León & Pérez-Mellado, 2016*). This transition in the trophic niche is particularly strong between spring (with the highest resource availability) and autumn (lowest), with hot summers marking a progressive limitation of resource availability (*Pérez-Cembranos, León & Pérez-Mellado, 2016*). According to a study based on individual fecal content analysis in NM and NG, the *P. lilfordi* seasonal response is sex specific (*Santamaría et al., 2019*), and particularly noticeable in autumn, when food resources are scarce and males show a despotic behavior (*Rotger, Igual & Tavecchia, 2020*), restricting the female niche amplitude (*Santamaría et al., 2019*). Despite a sex influence on both lizard metabolism and trophic niche behavior (*Pérez-Cembranos, León & Pérez-Mellado, 2016*), our results showed that males and females did not carry distinct gut microbiotas in any of the three islets, nor were microbial differences observed between sexes within a season (season-by-sex interaction was not significant for individual islets). Previous studies in lizards have shown both a significant and no sex effect on the gut microbiota (*Kohl et al., 2017*; *Zhang et al., 2022*), depending on the study system. In *P. lilfordi* gut microbiota a lack of sex effect could be associated to an omnivorous diet (with no clear sex-specific food preferences), a reduced sexual dimorphism (*Rotger, Igual & Tavecchia, 2020*) and a large microbial metacommunity effect within the discrete boundaries of an island (*Miller, Svanbäck & Bohannan, 2018*). Microbes can be largely transmissible in highly social or closed populations, due to the increased probability of contact among specimens (*Tung et al., 2015*; *Raulo et al., 2021*), allowing the rapid circulation of bacteria within the population. Furthermore, episodes of cannibalism are known within the genus *Podarcis* due to the restricted resources available in the islands (*Cooper, Dimopoulos & Pafilis, 2015*). Both sociality and cannibalism could provide a means of bacteria transmission between sexes, resulting in microbiota homogenization.

## Temporal dynamics of the gut microbiota within populations: persistence and seasonal fluctuations

We explored the short-term temporal stability and dynamics of the gut microbiota within populations. An important fraction of the microbiota (at least 30% of ASVs for each islet) persisted within a population across all four sampling events. As our study did not compare the same set of individuals over time, such persistence should be considered as the

maintenance of specific ASVs within the host population microbial metacommunity, not at individual level (*Robinson, Bohannan & Britton, 2019*). This persistent fraction of microbial diversity is most likely an underestimate given a possible failure in ASV sequencing for some of the specimens. At the same time, by considering only ASVs with at least 50% occurrence among specimens per date, we can largely exclude a relevant contamination with environmental microbes derived from diet. This is also in line with several studies spanning a wide range of animal taxa and consistently showing a neglectable contribution of environmental-derived bacteria to the stable microbiota core (*Costello et al., 2010*; *Kohl et al., 2017*). Although the diet-associated microbial content of Lilford's wall lizard has yet to be characterized, the lack of microbial sex-specific differences, whereas diet is largely sex-specific (*Santamaría et al., 2019*) further supports this observation. Nonetheless, future in-depth characterization of the environmental microbiota, including the phyllosphere, will provide a necessary confirmation.

Interestingly, the taxonomic profile of these persistent ASVs was highly congruent between islets (Fig. 7), and comprised several taxa previously identified as highly heritable in vertebrates, including the fermentative families Lachnospiraceae and Ruminococcaceae (*Grieneisen et al., 2021*) in wild baboons) and the genera *Oscillospira, Bacteroides, Odoribacter, Anaerotruncus* and *Coprobacillus* (*Kohl et al., 2017*) in lizards). The majority of these taxa represent important metabolic players (*Kohl et al., 2016*; *Kohl et al., 2017*; *Gophna, Konikoff & Nielsen, 2017*) and are known for their ability to degrade vegetable fibres (particularly the genera *Oscillospira* and *Bac teroides*) (*Gophna, Konikoff & Nielsen, 2017*; *Patnode et al., 2019*). While lizards from these islands predominantly consume arthropods, particularly insects, they are known to partly feed also on plant material (including seeds, nectar and pollen) (*Pérez-Cembranos, León & Pérez-Mellado, 2016*; *Santamaría et al., 2019*) as a derived adaptation to the limited local resources (*Van Damme, 1999*). Presence and persistence of these core fermentative bacteria would support a putative role of the gut microbiota in extending the lizard trophic niche amplitude towards the consumption of vegetable matter. This is an intriguing hypothesis that has been previously proposed for lizards (*Kohl et al., 2016*) and other vertebrate systems, including the giant panda (*Zhu et al., 2011*) and bonnethead sharks (*Leigh, Papastamatiou & German, 2021*), and that will require a target study on the metabolic contribution of gut microbes in *Podarcis*.

We finally looked at the microbiota compositional dynamics as a function of season (*i.e.,* seasonal plasticity). Recent studies in humans, wild great apes and mice have shown that the gut microbiota can be highly plastic, enabling the host to buffer seasonal changes in available resources, balancing nutritional needs and conferring dietary flexibility (*Maurice et al., 2015*; *Amato et al., 2015*; *Smits et al., 2017*; *Hicks et al., 2018*; *Baniel et al., 2021*; *Huang et al., 2022*). To date, this seasonal effect has only been marginally explored in reptiles (*Kohl et al., 2017*; *Alemany et al., 2022a*).

Our findings indicated a clear seasonal shift in the *P. lilfordi*'s gut microbiota configuration, which replicated along the two sampled years according to multiple core microbial subsets (Fig. 8A and Fig. S5). This microbiota seasonal covariation was largely comparable in the two sister populations studied, in line with their similar genetic

background (*Rotger et al., 2021*) and comparable trophic niches (Fig. 6B, but see also fecal content from (*Santamaría et al., 2019*). Most seasonal microbial markers were associated to taxa that persisted across all sampling dates, while largely fluctuating in relative abundance (Fig. 8B), with no major compositional turnover. These results are in line with recent findings in *P. siculus,* showing a gut microbiota plastic response to diet manipulation, which was essentially driven by quantitative changes (*Buglione et al., 2022*). Major players in the *P. lilfordi's* microbial plasticity were identified as members of fermentative families (mostly Ruminococcaceae and Lachnospiraceae, Fig. 8B). These same taxa have been repeatedly involved in seasonal microbial reconfiguration in mammals (*Maurice et al., 2015*; *Baniel et al., 2021*) suggesting that they might represent the more plastic component of the vertebrate microbiota in response to temporal dietary/physiological shifts.

Increasing literature is showing that the seasonal microbial plasticity might underpin a critical functional metabolic plasticity in response to changes in the host nutritional and physiological demands (*Alberdi et al., 2016*; *Huang et al., 2022*; *Buglione et al., 2022*). In particular, changes in gut microbiota fermentative ability have been associated to optimization of a plant-based diet, thermoregulation and overall maintenance of energy balance (*Maurice et al., 2015*; *Sommer et al., 2016*; *Hicks et al., 2018*; *Baniel et al., 2021*; *Guo et al., 2021*). Whereas causality cannot be inferred from our current data, an intriguing hypothesis is that such plasticity might provide these insular lizards with the ability to cope with metabolic stress in their constrained environments. Unlike mammals, reptiles are ectothermic and their metabolic requirements and putative energetic dependence on gut microbes might be under different regulatory processes (*Moeller et al., 2020*), a fascinating avenue that is worth further research.

## CONCLUSIONS

This study provides an in-depth exploration of the trends governing gut microbial dynamics between and within natural populations of *P. lilfordi*. By taking advantage of individual-based microbiota data and performing comparative analyses of three sister populations found in near islets, we showed that microbial diversity among populations is primarily driven by small qualitative changes, that is by the presence of few islet-specifics bacterial ASVs, with neglectable variation in major taxa membership. It remains unclear to what extent these small differences in community composition are adaptive, for instance in response to population adjustment to the trophic niche, or shaped by ecological drift, including a putative differential retention of ancestral taxa from the common ancestral population. Persistence of microbial taxa over time with no major compositional turnover support a strong resilience of these gut microbial communities along the short-term evolutionary times of their host diversification, implying strength and specificity of this symbiosis. A crucial aspect to clarify is to which extent these gut bacteria are transmitted among *Podarcis* individuals and through generations. Despite such compositional stability, replicated quantitative changes in microbial reconfiguration along seasons indicated that the microbiota is, to some extent, a plastic trait with a predictive temporal pattern in response to seasonal/dietary changes. The challenge is now to understand the impact of

microbial community composition and functional plasticity on the *P. lilfordi* fitness and ecological adaptation to these small islets, as well as to evaluate the great potential of integrative holobiont studies in monitoring this endangered wild species.

## ACKNOWLEDGEMENTS

We thank I. Hendriks for aid in sample collection.

### Funding

This study was supported by the Agencia Estatal de Investigación (AEI) and the Fondo Europeo de Desarrollo Regional (FEDER) (CGL2017-82986-C2-2-P) to L.B. and by the project PRD2018/25 (CAIB - Government of the Balearic Islands) to G.T. The funders had no role in study design, data collection and analysis, decision to publish, or preparation of the manuscript.

### Grant Disclosures

The following grant information was disclosed by the authors:
The Agencia Estatal de Investigación (AEI).
The Fondo Europeo de Desarrollo Regional (FEDER): CGL2017-82986-C2-2-P.
PRD2018/25 (CAIB - Government of the Balearic Islands).

### Competing Interests

The authors declare there are no competing interests.

### Author Contributions

- Laura Baldo conceived and designed the experiments, performed the experiments, analyzed the data, prepared figures and/or tables, authored or reviewed drafts of the article, and approved the final draft.
- Giacomo Tavecchia conceived and designed the experiments, performed the experiments, authored or reviewed drafts of the article, and approved the final draft.
- Andreu Rotger performed the experiments, authored or reviewed drafts of the article, and approved the final draft.
- José Manuel Igual performed the experiments, authored or reviewed drafts of the article, and approved the final draft.
- Joan Lluís Riera analyzed the data, prepared figures and/or tables, authored or reviewed drafts of the article, and approved the final draft.

### Field Study Permissions

The following information was supplied relating to field study approvals (i.e., approving body and any reference numbers):

Specimen manipulation and material sampling were carried out in accordance with the ethics guidelines and recommendations of the Species Protection Service (Department of Agriculture, Environment and Territory, Government of the Balearic Islands), under annual permits given to GT.

## DNA Deposition

The following information was supplied regarding the deposition of DNA sequences:

The original ASV abundance matrix per sample and corresponding taxonomic classification is available on Mendeley Data: Baldo, Laura (2022), "ASV_matrix_taxonomy_Plilfordi_microbiota", Mendeley Data, V1, doi: 10.17632/bc5nxsxgxd.1.

## Data Availability

The raw 16S rRNA Miseq Data is available at Bioproject: PRJNA764850.

## Supplemental Information

Supplemental information for this article can be found online at http://dx.doi.org/10.7717/peerj.14511#supplemental-information.

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
