# Peer review of "Insular holobionts: persistence and seasonal plasticity of the Balearic wall lizard (Podarcis lilfordi) gut microbiota"

_PeerJ, doi:10.7717/peerj.14511_

## Round 0.1 · original submission · Minor Revisions

Both reviewers have emphasized the high standard to which this work has been performed and the findings are presented, with which I wholeheartedly agree. Overall, the reviewers’ assessment of your work is favourable, and includes detailed suggestions that we hope will help shape a revision of your manuscript.

These suggestions include:
- Further clarification, additional detail, literature support (where applicable), or additional references for specific statements in the introduction and discussion;
- Streamlining of order of reporting between the methods and results section for consistency;
- More detail on sample sizes, analyses or statistical tools to be included in the results section;
- Specific suggestions to add additional analysis and/or adjust color palette in plots.

Reviewer 1 ·

Basic reporting

This is an excellent paper covering a key question in wild animal gut microbiota ecology and evolution. It calls on a range of data types to contextualize the microbial community analyses. I have no major concerns with either the analyses or the presentation of the paper, and my suggestions below are targeted toward presenting this paper to the broadest possible audience.

Introduction

Line 86: what have longitudinal studies revealed that single time-point studies can’t? Alternatively, what might they reveal that can’t be learned from single time points? (for example you reference buffering of seasonal energy needs on line 144 – that could be introduced here)

Lines 87-95: the claim that selection is reduced on islands on the surface seems at odds with the claim that they are ideal laboratories to study coevolution. Is there available evidence to suggest that reduced selection pressure from interspecific competition or predation on islands allows selection due to coevolution to become more detectable? Or os there some other argument to be made to reconcile the two statements?

Line 104: Can you provide an approximate date of common ancestry based on the literature?

Line 130: Since omnivory is fairly rare in small lizards, can you discuss the specifics here? For example, what parts of the plants to they feed on and on the seasonality of the plant feeding? This might be particularly useful in setting up the seasonality results in lines 337-345.

Methods

Line 260: Please explain why you choose to split the dataset into seasons here, then recombine the results later? Could a single season dominate the analysis without the splitting step, or is there another reason?

Results

Line 298: Were any of the island-specific core microbial taxa present across more than one island?

Line 328: typo – should the marginal year effect p value read 0.0396?

Discussion

Line 553: delete ‘Following’ from the sentence

Line 555: ‘sampling events’ rather than ‘sampling’

Line 583 or Line 604-614: I suggest a brief discussion of what is known about the gut microbiota of other largely carnivorous or insectivorous taxa that have shifted to a more omnivorous diet. For example, baleen whales (ie Sanders et al. 2015) or bonnethead sharks (ie Leigh et al. 2018), among others.

Line 587: I would add Amato et al. 2015 to the citation list (DOI 10.1007/s00248-014-0554-7)

Line 622: I’m not sure this claim follows from your data. While it is a possible interpretation, I would qualify the claim and/or suggest future work to confirm the hypothesis.

Figures

Figure 3, 4, 7: Are the color palettes in these figures colorblind friendly? They look like they might be, but if not the red and green might be difficult to distinguish.

Figure S2: ‘Phylum’ and ‘Family’ legend labels are identical.

Experimental design

Methods

Line 150: Can you provide the sample sizes per time point as well for each island?

Line 154: Can you provide the sample sizes for each time point?

Validity of the findings

All findings are valid and well presented. This paper is an important contribution to the field.

·

Basic reporting

“Insular holobionts: persistence and seasonal plasticity of the Balearic wall lizard (Podarcis lilfordi) gut microbiota (#77184)”

The study was conducted from 2017 to 2018 on three island lizard populations, during two seasons (spring and autumn) in order to identify the effect of islet, sex, lifestyle and season on the composition of the gut microbiota. The authors found microbiota diversity was strongly marked by seasonality with no sex effect and a marginal life stage and annual effect.
The work is well structured, with a satisfactory experimental design, relevant figures and well presented results. However, below are some comments that could improve the paper.
It is necessary to maintain consistency in the order between the chapter of methods and that of the results, for example the results of beta diversity appear in the text before those of alpha diversity, although the order is reversed in the methods. The same is true between the results shown in figure 5 and figure 6, they must also be mentioned in this order in the methods chapter.

Lines 524-527: It is recommended to cite also other references:
Raia, P., Guarino, F.M., Turano, M. et al. The blue lizard spandrel and the island syndrome. BMC Evol Biol 10, 289 (2010). https://doi.org/10.1186/1471-2148-10-289
Monti, D. M., Raia, P., Vroonen, J., Maselli, V., Van Damme, R., & Fulgione, D. (2013). Physiological change in an insular lizard population confirms the reversed island syndrome. Biological Journal of the Linnean Society, 108(1), 144-150

Experimental design

In order to identify Bacteria taxa driving differences in microbiota composition across populations I also suggest to use the ANCOM-II method and make a comparison with the other methods used.
Figure 3: provide the variance explained by the coordinates.
Beta and alpha diversity analyses: Did the authors do a pairwise analysis? Clarify it in the text.

Validity of the findings

Analyzing the results, the population/islet showing the greatest differences is EC, but looking at the number of samples making up the 3 populations, it can be seen that it is not always homogeneous, especially for the EC population, with only 19 samples. Is it likely that the differences that emerged were due to this element? Did the authors carry out a test to assess whether the number of samples in the three groups is suitable for the different analyzes?

Additional comments

Line 162: close the parenthesis after “old” and include the two references in a new parenthesis.
Line 212: add space after the parenthesis, and adjust the word “and”.

Line 428: cite appropriately the reference cited in the text.

Figure 3: I recommend adding letters to the panels of the figure and a specification in the caption that can help the reader in interpreting the results.

---

## Round 0.2 · accepted · Accept

Thank you for the thorough revision and congratulations on this excellent manuscript.

---

## Author Rebuttal · Round 0.2

Dear Editor,

Please find enclosed the revised version of the manuscript entitled "Insular holobionts: persistence and seasonal plasticity of the Balearic wall lizard (Podarcis lilfordi) gut microbiota".

We thank the reviewers for constructive suggestions and favorable overview of our study.

We have now addressed the minor concerns raised and modified the manuscript accordingly. Specifically, we have added further literature and additional clarifications to the Introduction and Discussion.

We streamlined the order of methods and results and changed the figures palette to make it color-blind friendly. We also added further details on sample sizes and clarified the statistical tools used, adding a visual representation of pairwise differences in figure 4.

As for the suggestion of reviewer 2 to estimate discriminatory features with an additional program (ANCOM-II), we considered this analysis was not necessary as our findings are already supported by two distinct and robust methods that were chosen after evaluating a set of different methods and considering sparsity for estimating presence/absence markers (but see specific comment for details).

We hope that the minor modifications on the manuscript have improved the overall reading and clarity of our findings and hope that the manuscript will be now suitable for publication.

Sincerely,

Laura Baldo, on the behalf of all the Authors

**Point-by-point detailed answers**

Editor

Both reviewers have emphasized the high standard to which this work has been performed and the findings are presented, with which I wholeheartedly agree. Overall, the reviewers' assessment of your work is favourable, and includes detailed suggestions that we hope will help shape a revision of your manuscript.

These suggestions include:

- Further clarification, additional detail, literature support (where applicable), or additional references for specific statements in the introduction and discussion;

- Streamlining of order of reporting between the methods and results section for consistency;

- More detail on sample sizes, analyses or statistical tools to be included in the results section;

- Specific suggestions to add additional analysis and/or adjust color palette in plots.

Reviewer 1 (Anonymous)

Basic reporting

This is an excellent paper covering a key question in wild animal gut microbiota ecology and evolution. It calls on a range of data types to contextualize the microbial community analyses. I have no major concerns with either the analyses or the presentation of the paper, and my suggestions below are targeted toward presenting this paper to the broadest possible audience.

A. Thank you, we really appreciate this comment.

**- Line 86: what have longitudinal studies revealed that single time-point studies can't? Alternatively, what might they reveal that can't be learned from single time points? (for example you reference buffering of seasonal energy needs on line 144 – that could be introduced here)**

A. We agree that this is an important point to stress, in terms of what temporal data of the gut microbiota can add to current understanding of microbial adaptive role or even just diversity. Following this suggestion, we briefly stated this importance in this paragraph, changing the text as follows:

"Studies of temporal variability of the gut microbiota are particularly critical to reach a comprehensive understanding of the diversity of these communities, as well as to shed light into their adaptive potential. Recent long-term studies in vertebrates have shown that the gut microbiota can be highly plastic (Alberdi et al., 2016; Gomez et al., 2019; Buglione et al., 2022), with seasonal fluctuations in response to the host's physiological adjustments and dietary changes over time (Amato et al., 2015; Smits et al., 2017; Hicks et al., 2018; Baniel et al., 2021; Guo et al., 2021; Huang et al., 2022). This suggests a microbial role in buffering the host metabolic needs, which can effectively boost its ecological adaptation by an increase in phenotypic plasticity (Huang et al., 2022). Yet, unlike single time point studies, long-term population-level studies of gut microbiota in wild animals remain particularly rare, due to inherent difficulties in individual data collection and demographic monitoring."

- **Lines 87-95: the claim that selection is reduced on islands on the surface seems at odds with the claim that they are ideal laboratories to study coevolution. Is there available evidence to suggest that reduced selection pressure from interspecific competition or predation on islands allows selection due to coevolution to become more detectable? Or os there some other argument to be made to reconcile the two statements?**

A. This is an interesting discussing point. Selection in islands is known to be relaxed at enemies' interactions (predators) and interspecific competition (two major selective pressures in open populations). At the same time, insular populations experience increased intraspecific competition, which favors local adaptation and the expansion of the trophic niche while promoting a slower life cycle over time (low fecundity and high survival, k-strategy). We agree with the referee that currently there is no evidence that reduced interspecific interactions would make the coevolution more detectable, although this is an interesting hypothesis. We therefore decided to reframe the paragraph by pointing out the lack of current studies of microbial symbiosis in island systems.

-**Line 104: Can you provide an approximate date of common ancestry based on the literature?**

A. We now added the approximate date of common ancestry, estimated to be 12000 years, according to literature

-**Line 130: Since omnivory is fairly rare in small lizards, can you discuss the specifics here? For example, what parts of the plants to they feed on and on the seasonality of the plant feeding? This might be particularly useful in setting up the seasonality results in lines 337-345.**

A. A dietary shift towards an omnivory is typical of lizards in islands where local resources are scarce and intraspecific competition is high, leading to a wider trophic niche (and partial herbivory). In these islands in particular, fecal content studies have shown that these lizards feed on a wide range of dietary items, especially arthropods (particularly ants), but also

mollusks and plant pollen, nectar and seeds. Plant consumption appears to integrate the diet during summer and autumn seasons, when other food sources are scarce. We now added this info to the introduction section with appropriate references.

**-Line 260: Please explain why you choose to split the dataset into seasons here, then recombine the results later? Could a single season dominate the analysis without the splitting step, or is there another reason?**

A. The dataset was split according to "season" as this was a highly significant variable in structuring the microbiota (according to PERMANOVA). To assess only the effect of "islet" in driving microbiota diversity, we decided to work on individual seasonal datasets (either spring or autumn). This allowed to retrieve only markers that were "islet-specific" but "season-independent". We now further clarified this point in the methods.

**- Line 298: Were any of the island-specific core microbial taxa present across more than one island?**

A. Of the ASVs with 80% frequency within each islet ("islet core"), 24 were present in all three islets ("common core). This result was now added to the Results section, discriminating between the two types of cores.

**-Line 328: typo – should the marginal year effect p value read 0.0396?**

A. Correct. We revised the value.

**-Line 553: delete 'Following' from the sentence**

A. Done

**-Line 555: 'sampling events' rather than 'sampling'**

 A. Done

**-Line 583 or Line 604-614: I suggest a brief discussion of what is known about the gut microbiota of other largely carnivorous or insectivorous taxa that have shifted to a more omnivorous diet. For example, baleen whales (ie Sanders et al. 2015) or bonnethead sharks (ie Leigh et al. 2018), among others.**

A. Thanks for the suggestion. We added a short reference to other lizards, panda and bonnethead sharks that show a transition towards herbivory.

-**Line 587: I would add Amato et al. 2015 to the citation list (DOI 10.1007/s00248-014-0554-7)**

A. Thanks for the useful suggestion. We read this interesting study with strong parallelisms with our findings and cited it in the introduction and discussion. We also added a recent relevant citation by Huang et al. 2022 (Cell Reports) showing a critical role of seasonal shift in gut microbiota for regulating the circadian rhytmn in the giant panda.

-**Line 622: I'm not sure this claim follows from your data. While it is a possible interpretation, I would qualify the claim and/or suggest future work to confirm the hypothesis.**

A. Following this suggestion, we decided to remove this short paragraph as too speculative.

-**Figure 3, 4, 7: Are the color palettes in these figures colorblind friendly? They look like they might be, but if not the red and green might be difficult to distinguish.**

A. We now changed the three figures, including Figure 8 to a colorblind friendly palette (the Okabe-Ito palette).

-**Figure S2: 'Phylum' and 'Family' legend labels are identical.**

A. Figure 2 was now revised and correctly labelled.

-**Line 150: Can you provide the sample sizes per time point as well for each island?**

A. Sample size for time point and islet (10 estimates overall) can be accessed in Table S1.

-**Line 154: Can you provide the sample sizes for each time point?**

A. The sample sizes were now added to the text.

All findings are valid and well presented. This paper is an important contribution to the field.

Reviewer 2 (Domenico Fulgione)

Basic reporting

"Insular holobionts: persistence and seasonal plasticity of the Balearic wall lizard (Podarcis lilfordi) gut microbiota (#77184)"

The study was conducted from 2017 to 2018 on three island lizard populations, during two seasons (spring and autumn) in order to identify the effect of islet, sex, lifestyle and season on the composition of the gut microbiota. The authors found microbiota diversity was strongly marked by seasonality with no sex effect and a marginal life stage and annual effect.

The work is well structured, with a satisfactory experimental design, relevant figures and well presented results. However, below are some comments that could improve the paper.

**-It is necessary to maintain consistency in the order between the chapter of methods and that of the results, for example the results of beta diversity appear in the text before those of alpha diversity, although the order is reversed in the methods. The same is true between the results shown in figure 5 and figure 6, they must also be mentioned in this order in the methods chapter.**

A. Thanks for the suggestion. We now changed the order of the methods chapters to match that of the results.

*-* **Lines 524-527: It is recommended to cite also other references:**

**Raia, P., Guarino, F.M., Turano, M. et al. The blue lizard spandrel and the island syndrome. BMC Evol Biol 10, 289 (2010). https://doi.org/10.1186/1471-2148-10-289**

**Monti, D. M., Raia, P., Vroonen, J., Maselli, V., Van Damme, R., & Fulgione, D. (2013). Physiological change in an insular lizard population confirms the reversed island syndrome. Biological Journal of the Linnean Society, 108(1), 144-150**

A. We thank the reviewer for the suggestion and the very interesting articles. However, we consider that these articles explore a topic, the reversed island syndrome, that was not discussed in our study and were therefore not appropriate for a citation. This discussion section focuses on the trophic behavior in populations that show the typical island syndrome, with no evidence of aggressiveness.

Experimental design

*-* **In order to identify Bacteria taxa driving differences in microbiota composition across populations I also suggest to use the ANCOM-II method and make a comparison with the other methods used.**

A. Thanks for the suggestion. We currently estimated the discriminatory taxa across islets and seasons using a double approach (Lefse and indval) and crossed results to be conservative and exclude false positives. Before selecting these two methods we tested several options currently available for enrichment studies of microbiome, including Deseq2, corncob and Ancom, among others. Several of these methods do not account for clear

presence/absence of taxa, while detecting only features that are present over time but change in quantitative or expression levels. As microbiome datasets are highly sparse (zero-inflated) and because we were interested in also detecting markers that uniquely occurred in one island or season (presence/absence), we decided to use the two most robust methods that fit our goal, indval and LeFse. We are aware that there is a plethora of more methods available (see Nearing et al. 2022, Nature Communications), however we believe this double approach was a fair compromise, met our goals and provided us with conservative and robust findings. We now tried to further clarify this point in the methods.

-**Figure 3: provide the variance explained by the coordinates.**

A. We now added the variance explained in Figure 3.

- **Beta and alpha diversity analyses: Did the authors do a pairwise analysis? Clarify it in the text.**

A. A Post-hoc tests were performed after PERMANOVA to test for a season effect by islet, as indicated in the results section (lines 353-355). Posthoc tests were needed due to the presence of a significant interaction between season and islet, which mean that seasonal effects differ in size by islet –but are significant in all cases. Pairwise comparisons were also performed for alfa diversity measures after ANOVA, as explained in the results sections. We have made this clearer by adding this information to the methods sections and redoing Figure 4 to include letter codes to indicate differences among samples. We have also redone the figure with islet order and color for seasons to emphasize how differences among islets occur in autumn, but not in spring.

Validity of the findings

- **Analyzing the results, the population/islet showing the greatest differences is EC, but looking at the number of samples making up the 3 populations, it can be seen that it is not always homogeneous, especially for the EC population, with only 19 samples. Is it likely that the differences that emerged were due to this element? Did the authors carry out a test to assess whether the number of samples in the three groups is suitable for the different analyzes?**

A. Unbalanced designs are often trickier to interpret, but we have exerted extra caution in doing so. ANOVA is quite robust to small unbalances, especially when variance is homoscedastic (which was the case). In addition, for both PERMANOVA (beta diversity) and ANOVA (alpha diversity), differences in the contrasts of interest were highly significant, they were apparent in graphical depictions, and we used marginal Type II tests to ensure correct contrasts.

Additional comments

**-Line 162: close the parenthesis after "old" and include the two references in a new parenthesis.**

A. Done

**-Line 212: add space after the parenthesis, and adjust the word "and".**

A. Done

**-Line 428: cite appropriately the reference cited in the text.**

A. revised

**-Figure 3: I recommend adding letters to the panels of the figure and a specification in the caption that can help the reader in interpreting the results.**

A. The figure and legend were changed as suggested.